# AT2 Receptor Mediated Activation of the Tyrosine Phosphatase PTP1B Blocks Caveolin-1 Enhanced Migration, Invasion and Metastasis of Cancer Cells

**DOI:** 10.3390/cancers11091299

**Published:** 2019-09-03

**Authors:** Samuel Martínez-Meza, Jorge Díaz, Alejandra Sandoval-Bórquez, Manuel Valenzuela-Valderrama, Natalia Díaz-Valdivia, Victoria Rojas-Celis, Pamela Contreras, Ricardo Huilcaman, María Paz Ocaranza, Mario Chiong, Lisette Leyton, Sergio Lavandero, Andrew F.G. Quest

**Affiliations:** 1Advanced Center for Chronic Diseases (ACCDiS), Faculty of Chemical & Pharmaceutical Sciences and Faculty of Medicine, University of Chile, Santiago 8380000, Chile (S.M.-M.) (J.D.) (A.S.-B.) (M.V.-V.) (N.D.-V.) (V.R.-C.) (P.C.) (R.H.) (M.C.) (L.L.); 2Center for Studies on Exercise, Metabolism and Cancer (CEMC), Institute of Biomedical Sciences (ICBM), Faculty of Medicine, University of Chile, Santiago 8380000, Chile; 3Instituto de Innovación e Investigación en Salud, Facultad de Ciencias de la Salud, Universidad Central de Chile, Santiago 8320000, Chile; 4Division of Cardiovascular Diseases, Advanced Center for Chronic Diseases (ACCDiS), Facultad de medicina, Pontificia Universidad Catolica de Chile, Santiago 8330024, Chile; 5Cardiology Division, Department of Internal Medicine, University of Texas Southwestern Medical Center, Dallas, TX 75390, USA; 6Corporación Centro de Estudios Científicos de las Enfermedades Crónicas (CECEC), Santiago 7860201, Chile

**Keywords:** AT2 receptor, Caveolin-1, PTP1B, metastasis

## Abstract

The renin–angiotensin receptor AT2R controls systemic blood pressure and is also suggested to modulate metastasis of cancer cells. However, in the latter case, the mechanisms involved downstream of AT2R remain to be defined. We recently described a novel Caveolin-1(CAV1)/Ras-related protein 5A (Rab5)/Ras-related C3 botulinum toxin substrate 1 (Rac1) signaling axis that promotes metastasis in melanoma, colon, and breast cancer cells. Here, we evaluated whether the anti-metastatic effect of AT2R is connected to inhibition of this pathway. We found that murine melanoma B16F10 cells expressed AT2R, while MDA-MB-231 human breast cancer cells did not. AT2R activation blocked migration, transendothelial migration, and metastasis of B16F10(cav-1) cells, and this effect was lost when AT2R was silenced. Additionally, AT2R activation reduced transendothelial migration of A375 human melanoma cells expressing CAV1. The relevance of AT2R was further underscored by showing that overexpression of the AT2R in MDA-MB-231 cells decreased migration. Moreover, AT2R activation increased non-receptor protein tyrosine phosphatase 1B (PTP1B) activity, decreased phosphorylation of CAV1 on tyrosine-14 as well as Rab5/Rac1 activity, and reduced lung metastasis of B16F10(cav-1) cells in C57BL/6 mice. Thus, AT2R activation reduces migration, invasion, and metastasis of cancer cells by PTP1B-mediated CAV1 dephosphorylation and inhibition of the CAV1/Rab5/Rac-1 pathway. In doing so, these observations open up interesting, novel therapeutic opportunities to treat metastatic cancer disease.

## 1. Introduction

The renin–angiotensin system plays a fundamental role in controlling vasoconstriction, cardiovascular function, and renal homeostasis [1]. Peptides called angiotensins are the endogenous ligands for two of the relevant seven-transmembrane spanning G protein-coupled receptors, namely type 1 (AT1R) and type 2 (AT2R) receptors. AT1R activation increases blood pressure and cell proliferation by enhancing the activity of kinases in the mitogen activated protein kinase (MAPK)/extracellular signal regulated kinase (ERK), phosphatidyl inositol-3-kinase (PI3K)/protein kinase B (PKB), and Janus kinase (JAK)/signal transducer and activator of transcription (STAT) pathways [2,3]. Conversely, the antagonistic AT2R prevents these effects, mainly by activating phosphatases, such as SH2-containing protein tyrosine phosphatase-1 (SHP-1), MAPK phosphatase-1 (MKP-1), and protein phosphatase 2A (PP2A), which dephosphorylate effector proteins downstream of AT1R [4].

Expression of AT1R is fairly ubiquitous, while AT2R is mainly expressed during embryonic development and decreases in adult organisms [5]. However, in some pathological conditions, like cancer, both receptors are overexpressed [6] and participate in the control of cancer development [7]. AT1R is implicated in promoting metastasis [8], whereas AT2R decreases metastasis in colorectal, breast, and prostate cancer [9,10,11]. Taken together, this evidence posits AT2R as a promising target for the treatment of cancer [12], yet the mechanisms by which this receptor elicits responses in cancer cells remains largely unknown.

Among the many molecular mechanisms that contribute to metastasis, our group and others have shown that increased expression of the scaffolding protein caveolin-1 (CAV1) is associated with cancer progression [13,14]. More recent studies from our group identified a novel CAV1/Ras-related protein 5A (Rab5)/Ras-related C3 botulinum toxin substrate 1 (Rac1) signaling axis that is activated when CAV1 is phosphorylated on tyrosine-14 (Y14) [15,16,17]. CAV1 is reportedly phosphorylated by nonreceptor tyrosine kinases, like Src, Fyn, Yes, and Abl [13]. Alternatively, phosphatases, such as the nonreceptor protein tyrosine phosphatase type 1B (PTP1B), reportedly dephosphorylate (pY14)CAV1 [18]. This phosphatase is a well-known regulator of the insulin signaling pathway [19]; however, in the context of cancer its role is controversial, since it reportedly can either promote or prevent the development of cancer [20]. Interestingly, the renin–angiotensin system has been connected previously to CAV1 phosphorylation. For instance, AT1R was shown to increase CAV1 phosphorylation on Y14 and promote epithelial–mesenchymal transition in kidney cells [21]. However, although AT2R activation leads to dephosphorylation of many AT1R effector molecules, CAV1 as a target of AT2R signaling, and the possible consequences thereof in the context of cancer, have not been described.

Here, we show that AT2R activation decreases CAV1-enhanced melanoma and breast cancer migration and invasion. This ability of AT2R is linked to activation of the tyrosine phosphatase PTP1B, dephosphorylation of (pY14)CAV1, and inhibition of the CAV1/Rab5/Rac-1 signaling axis. Finally, AT2R activation was also shown to block CAV1-enhanced melanoma metastasis in a preclinical animal model. In summary, the observations reported here yield novel insights to the mechanisms by which AT2R inhibits cancer cell metastasis and, in doing so, open up unanticipated opportunities that can be therapeutically exploited to prevent metastatic disease, the leading cause of deaths in cancer patients.

## 2. Results 

### 2.1. AT2R Is Differentially Expressed in Melanoma and Breast Cancer Cell Lines, and Its Activation Reduces Migration

We evaluated by western blotting AT2R expression in B16F10 murine melanoma and MDA-MB-231 human breast cancer cells. Both cell lines, expressing CAV1 or not, were previously characterized by our laboratory [15,22]. B16F10 cells were either transfected with the empty vector (placIOP) alone or the vector containing a CAV1-encoding insert (placIOP-cav-1) to generate the cell lines B16F10(mock) and B16F10(cav-1), respectively. Alternatively, MDA-MB-231 were either transduced with control or cav-1-specific short-hairpin RNAs (shRNAs) to generate MDA-MB-231(sh-ctrl) and MDA-MB-231(sh-cav-1) cells, respectively. B16F10(mock) and B16F10(cav-1) express similar levels of AT2R, suggesting that AT2R expression is independent of CAV1 presence. Alternatively, AT2R was not detected in MDA-MB-231(sh-ctrl) or MDA-MB-231(sh-cav-1) cells (Figure 1a). 

To evaluate whether AT2R activation modulated B16F10 cell behavior, we stimulated cells with the specific synthetic AT2R agonist N-α−Nicotinoyl-Tyr-Lys-(N-α-Z-Arg)-His-Pro-Ile (CGP42112) [23]. Our results indicated that B16F10 migration decreased significantly when cells were preincubated with CGP42112 (1 µM), but this effect was only observed in B16F10(cav-1) cells (Figure 1b). In contrast, stimulation of MDA-MB-231 with CGP42112 did not alter their migration (Figure 1c), as was to be expected due to the absence of AT2R in these cells (Figure 1a). To corroborate these results, we also employed Ang II and Ang-(1–9), two known endogenous agonists of AT2R. In both cases, a decrease in the migration of B16F10(cav-1) cells was observed (Appendix A). For subsequent experiments, we used CGP42112, which is more specific and thought to exclusively activate AT2R.

To corroborate these findings, AT2R was silenced in B16F10(cav-1) cells using an shRNA specific for AT2R to yield B16F10(cav-1/sh-AT2R) cells. In subsequent experiments, cells transfected with a scrambled shRNA construct, B16F10(cav-1/sh-ctrl), were used as controls. AT2R expression decreased in B16F10(cav-1/sh-AT2R) clones 2 (C2) and 5 (C5) cells, but only in the latter case was the decrease significant with respect to B16F10(cav-1/sh-scramble) cells. Of note, a modest decline in AT2R expression was also observed in B16F10(cav-1/sh-ctrl) cells (Figure 2a).

The results obtained in a transwell assay indicated that B16F10(cav-1/sh-AT2R) cell migration was elevated in comparison with B16F10(cav-1/sh-ctrl) cells. Also, B16F10 (cav-1/sh-AT2R) cells did not respond to CGP42112 stimulation, while the control cells did (Figure 2b), confirming the previous observation (Figure 1b) that AT2R activation suppressed B16F10(cav-1) cell migration.

Then, we examined the effect of AT2R on CAV1-dependent migration in another cell line. MDA-MB-231 cells express high endogenous levels of CAV1 (Figure 1a), but they lack the AT2R. So, we overexpressed AT2R in these cells using an adenovirus encoding AT2R (AdNHA-AT2R), which contained a bicistronic vector that permits co-expression of both the angiotensin receptor coupled to an HA-tag and the green fluorescent protein (GFP) [24]. Adenovirus with a vector for GFP only was used as a control (Ad ctrl). Transduction of MDA-MB-231 with AdNHA-AT2R or Ad-ctrl was verified by flow cytometry analysis, and AT2R presence was corroborated by both western blotting using antibodies specific for the HA-tag (Figure 2c) and flow cytometry (Appendix A). The western blot image shown was cropped from the original data shown in (Appendix A). Results obtained in the transwell migration assay indicated that AT2R overexpression only decreased migration of cells that endogenously expressed CAV1 (MDA-MB-231 sh-ctrl), but not of the CAV1 silenced cells (MDA-MB-231 sh-cav-1). Rather surprisingly, this effect was seen even in absence of the AT2R agonist CGP42112 (Figure 2d). Collectively, the data obtained in different cell lines coincided in showing that AT2R activation/stimulation reduced CAV1-enhanced migration.

### 2.2. AT2R Decreases CAV1/Rab5/Rac1 Activity in Melanoma Cells 

CAV1 promotes cell migration/invasion in a manner dependent on Y14 phosphorylation (pY14)CAV1. Thus, we evaluated (pY14)CAV1 levels in B16F10 cells stimulated with CGP42112. Western blot analysis indicated that CGP42112 significantly decreased (pY14)CAV1 levels after 30 min of incubation (Figure 3a). To confirm that AT2R promoted CAV1 dephosphorylation, we also evaluated (pY14)CAV1 levels in AT2R-silenced B16F10 cells and observed that reduced AT2R expression correlated with an increase in CAV1 phosphorylation levels (Figure 3b). Additionally, we evaluated (pY14)CAV1 levels in MDA-MB-231 cells overexpressing AT2R. While a slight decrease was observed (Appendix A), this reduction was not statistically significant, presumably because the MDA-MB-231 cells express very high levels of CAV1.

To further evaluate how AT2R reduced migration, we next tested the effect of CGP42112 in cells expressing the phosphomimetic mutant CAV1(Y14E) of CAV1. As expected, CGP42112 did not decrease migration of B16F10 cells expressing CAV1(Y14E), confirming that AT2R inhibits migration by decreasing (pY14)CAV1 (Figure 3c). As a control, CAV1 levels were determined by western blotting in cells expressing wild-type CAV1 versus CAV1(Y14E), and similar levels were observed in both cases (Appendix A).

At this point, the available evidence indicated that activation of AT2R inhibited the CAV1/Rab5/Rac1 pathway at the CAV1 level, suggesting that downstream components should also decrease their activity. To evaluate this possibility, we measured the GTP loading of Rab5 and Rac-1 in pull-down assays. First, we evaluated the Rab5 activity in B16F10 cells preincubated or not with CGP42112 (Figure 3d). Indeed, we observed a tendency towards reduced Rab5 activation in B16F10(cav-1) but not in B16F10(mock) cells. Although not significant by ANOVA analysis (see Figure 3d), the comparison of B16F10(cav-1) cells treated or not with CGP42112 in a Student’s *t*-test did yield a significant difference (*p* ≤ 0.05; not shown). Likewise, Rac1 activation also decreased only in CPG42112-treated B16F10(cav-1) cells (Figure 3e). Taken together, these results suggest that AT2R diminishes CAV1-enhanced GTP loading of Rab5 and Rac1.

### 2.3. AT2R Inhibits the CAV1/Rab5/Rac-1 Pathway through PTP1B Activation

Results from the literature have implicated PTP1B as a phosphatase that dephosphorylates Y14 [18], and AT2R is known to activate phosphatases. Thus, to determine if PTP1B inhibited the CAV1/Rab5/Rac1 pathway, we evaluated the effect of the PTP1B inhibitor Cinn-GEL2-ME on AT2R-induced (pY14)CAV1 dephosphorylation and cell migration. We observed that Cinn-GEL2-ME not only reduced AT2R-induced (pY14)CAV1 dephosphorylation (Figure 4a) but also increased migration, even in the absence of CGP42112 (Figure 4b). As a control, to discard Src kinase as a direct target of AT2R, phosphorylation of Src associated with kinase activation was evaluated and found to be unchanged in the presence of CGP42112 (Appendix A).

Several reports implicate AT2R as a phosphatase activator [4,6]. To determine whether AT2R stimulation increases PTP1B activity relevant to CAV1 dephosphorylation, we evaluated phosphatase activity in CAV1 immunoprecipitates (IPs) from B16F10(cav-1) cells. We observed that the activity was higher in CAV1 IPs from B16F10(cav-1) cells stimulated with CGP42112, and this increase was blocked if cells were treated with the inhibitor Cinn-GEL2-ME (Figure 4c). Western blot analysis of CAV1 IPs confirmed that the increase in phosphatase activity was not attributable to increases in phosphatase protein presence after CGP42112 treatment (Appendix A). Together, these results show that AT2R stimulation increases PTP1B phosphatase activity, which, in turn, dephosphorylates CAV1 and thereby decreases the migration of B16F10(cav-1) cells.

Previous results from our group have shown that the CAV1 phosphorylation on Y14 is required for CAV1 to enhance extravasation [15,17]. Here we evaluated the effect of AT2R stimulation on transmigration of B16F10 cells through an endothelial cell monolayer. Monolayer formation of endothelial cells was corroborated by evaluating the permeability to Dextran Blue (Appendix A). Indeed, as we anticipated based on the results in migration assays, stimulation with CPG42112 decreased transendothelial migration observed for B16F10(cav-1), but not B16F10(mock) cells (Figure 5a). The role of AT2R in this process was confirmed by showing that CPG42112 treatment of the human melanoma cell line A375(cav-1) also decreased transendothelial migration (Figure 5b).

Finally, we corroborated the importance of AT2R in limiting CAV1-enhanced metastasis in vivo. B16F10 melanoma cells, expressing CAV1 or not, and AT2R were injected intravenously, after pretreating with either vehicle or CGP42112, and lung tumor mass was determined 21 d later (Figure 5c). Consistent with our previous results showing that CAV1 expression promotes B16F10 metastasis, lung tumor mass was higher for B16F10(cav-1) than B16F10(mock) cells. Importantly, AT2R stimulation with CGP42112 essentially blocked B16F10(cav-1) lung metastasis (Figure 5d). Taken together, these observations show that AT2R stimulation prevents the metastasis of B16F10(cav-1) cells. 

## 3. Discussion

Metastasis is responsible for approximately 90% of cancer-associated deaths [25], and, for that reason, it is becoming ever more necessary to develop treatments that block or at least delay this process. Renin–angiotensin system components have been implicated in cancer development. Indeed, receptors, such as AT1R, increase in expression and favor cancer progression. Surprisingly, antagonistic receptors, including AT2R, also increase is expression, at least at the onset, likely as a compensatory response [6]. How AT2R precludes cancer progression, and particularly metastasis, remains largely unknown [9,10,11]. Previous studies from our laboratory showed that CAV1 enhanced migration and invasion of melanoma, breast, and colorectal cancer cells [22,26]. Thus, we focused here on elucidating whether AT2R inhibits metastasis and how this may relate to the presence of CAV1. In this study, we identified a mechanism that explains the anti-metastatic role of AT2R in melanoma and breast cancer cells. 

AT2R expression can vary dramatically between tissues and amongst cell lines. AT2R is reportedly not expressed by normal melanocytes [27] but has been detected in human melanoma cell lines [28], consistent with our observations revealing that AT2R was present in mouse B16F10 melanoma cells. These findings are consistent with the notion that, during transformation of normal melanocytes to melanoma cells, AT2R expression is induced. The same has been reported in several types of cancers [6], including breast cancer, where the analysis of tumors from patients revealed a gradual increase in AT2R expression associated with progression to the metastatic stage [29]. Also, AT2R expression promotes migration of pancreatic adenocarcinoma cells [30] and stem cells [31]. However, as observed by others in breast cancer cells [32], we were unable to detect AT2R in the metastatic human breast cancer line MDA-MB-231. These discrepancies may simply reflect variations between cell lines and/or the stages of cancer development they represent. In any case, the availability of B16F10 cells expressing AT2R and MDA-MB-231 cells lacking the receptor provided unique opportunities to evaluate how knock-down (B16F10) or over expression (MDA-MB-231) of AT2R modulated CAV1-enhanced migratory and invasive behaviors in these cells.

Recently, AT2R activation was shown to increase mesenchymal stem cell migration by signaling through the focal adhesion kinase (FAK) and ras homolog family member A (RhoA)/cell division control protein 42 homolog (Cdc42) pathways [31]. A related mechanism has been uncovered by our group to explain how CAV1 promotes migration and invasion, not only in cancer, but also in dendritic cells [33]. Specifically, CAV1 was shown to activate a novel CAV1/Rab5/Rac-1 signaling axis that requires phosphorylation of CAV1 on tyrosine-14, which is also essential in B16F10 melanomas to promote lung metastasis [15,16,17]. The decrease in (pY14)CAV1 levels after incubation with CGP42112, which reportedly only activates AT2R [23], indicated that AT2R likely inhibited the CAV1/Rab5/Rac-1 pathway by diminishing (pY14)CAV1 dephosphorylation. However, not all CAV1 was dephosphorylated, suggesting that only a minor fraction of all CAV1 might be responsible for promoting migration. This interpretation is supported by our previous studies indicating that only a small fraction of all CAV1 translocates to focal adhesions (FAs) and is likely responsible there for promoting FA turnover and migration [17,34]. The analysis of B16F10 cells expressing a phosphomimetic CAV1 mutant (CAV1(Y14E)) support this conclusion because enhanced migration of such cells due to the expression of CAV1(Y14E) was not suppressed by CGP42112. 

The phosphorylation of CAV1 on Y14 is due to the nonreceptor tyrosine kinases Src, Fyn, and Abl that are activated under a wide variety of conditions and particularly during migration [13,14]. Alternatively, PTP1B is known to dephosphorylate (pY14)CAV1 [18] and is reportedly exquisitely sensitive to reactive oxygen species (ROS) levels [35,36]. In this context, PTP1B emerged as a good candidate because AT2R activation is known to reduce ROS levels via inhibition of nicotinamide adenine dinucleotide phosphate hydrogen (NADPH) oxidase [6], which likely promotes phosphatase activation. Moreover, PTP1B co-immunoprecipitated with CAV1, and activity of the phosphatase in such immunoprecipitates increased following AT2R activation. Taken together, this evidence supports a model whereby AT2R stimulation activates PTP1B, which dephosphorylates CAV1 Y14 and thereby prevents CAV1-enhanced migration. 

Consistent with our model that AT2R inhibits the CAV1/Rab5/Rac-1 pathway, we also provide evidence here showing that AT2R activation by CGP42112 decreases Rab5 and Rac-1 activation, and does so within 30 min, suggesting a post-translational mechanism. Nevertheless, potential alternative interpretations include reports showing that AT2R activation increases the expression of p85α [37], a Rab5 GTPase activating protein (GAP) that is sequestered by (pY14)CAV1 to enhance Rab5 activity downstream of CAV1 in the CAV1/Rab5/Rac-1 axis. In this scenario, AT2R would decrease Rab5 activity by a transcriptional mechanism involving increased p85α expression.

Alternatively, AT2R signaling might impact directly on Rac-1, via activation of PP2A, a serine/threonine phosphatase known to be activated downstream of AT2R. Although the precise role of PP2A remains unknown in this context, PP2A reportedly inhibits Rac-1 by preventing activation of T-lymphoma invasion and metastasis-inducing protein-1 (TIAM1) [38] and Trio [39], two guanine nucleotide exchange factors (GEFs) that augment Rac-1 activity.

CAV1 has been suggested to promote melanoma metastasis by enhancing extravasation [40]. Consistent with the latter findings, we have shown that CAV1 promotes transendothelial migration in vitro [29] and very early events in lung colonization by melanoma cells [41]. Indeed, we observed here that CAV1-enhanced transendothelial migration of melanoma cells is blocked by CGP42112 treatment. Most importantly, a significant decrease in lung metastasis was observed for B16F10(cav-1) cells treated with CGP42112.

Another intriguing finding here was that silencing alone of AT2R increased migration and (pY14)CAV1 levels, while overexpression decreased migration, even in the absence of CGP42112. Such agonist/ligand-independent activity of AT2R may be attributable to spontaneous homo-dimerization [42]. The activation of AT2R in the absence of ligand stimulation has been described in prostate cancer cells as well [43]. Such behavior of the AT2R needs to be taken into consideration when evaluating the potential of AT2R-based anti-metastatic therapies.

In summary (see model, Figure 6), the present study identified AT2R, an important receptor of the renin–angiotensin system, as a negative regulator of CAV1-enhanced migration, invasion, and metastasis and attributed this ability to PTP1B activation.

## 4. Materials and Methods

### 4.1. Reagents

Mouse monoclonal anti-caveolin-1 and mouse monoclonal anti-Rac1 antibodies were from Transduction Laboratories (Lexington, KY, USA). Rabbit polyclonal anti-actin was from R&D Systems (Minneapolis, MN, USA). Mouse monoclonal anti-Rab5, Rabbit Polyclonal anti-AT1R, and AT2R antibodies were from Santa Cruz Biotechnology (Santa Cruz, CA, USA). Goat anti-rabbit and goat anti-mouse immunoglobulin G (IgG) antibodies coupled to horseradish peroxidase were from Bio-Rad Laboratories (Hercules, CA, USA). Cell medium and antibiotics were from GIBCO Life Technologies (Grand Island, NY, USA). Fetal bovine serum (FBS) was from HyClone Laboratories (Logan, UT, USA). Glutathione-Sepharose 4B was from GE Healthcare (Piscataway, NJ, USA). The chemiluminescent substrate (EZ-ECL) and protein A/G beads were from Pierce Chemical (Rockford, IL, USA). The AT2R lentiviral short hairpin RNAs (shRNA) and the PTP1B inhibitor Cinn-gel-2ME were from Santa Cruz Biotechnology (Santa Cruz, CA, USA). Human fibronectin was from Becton Dickinson (San Jose, CA, USA). The AT2R lentiviral expression vector was kindly provided by Prof. Walter G. Thomas (University of Queensland, Brisbane, Australia). The soluble alkaline phosphatase substrate *p*-Nitro phenyl Phosphate (*p*NPP) was from Santa Cruz, CA, USA.

### 4.2. Cell Models and Cell Culture

Sublines obtained from the metastatic murine melanoma cell line B16F10 (B16F10(cav-1) and B16F10(mock)), or human melanoma cell line A375 (A375(cav-1) and A375(mock)), were previously described [17,22]. These cells were cultured in Roswell Park Memorial Institute (RPMI) medium with 10% FBS (10 μg/mL streptomycin and 10,000 U/mL penicillin) and maintained at 37 °C in a humid atmosphere with 5% CO_2_. Also, the sublines derived from the human metastatic breast cancer cell line MDA-MB-231, MDA-MB-231(sh-cav-1), and MDA-MB-231(sh-ctrl) have been previously described [15]. These cell lines were cultured in Dulbecco’s Modified Eagle Medium (DMEM) F12 (GIBCO Life Technologies (Grand Island, NY, USA) with 10% FBS, 1% penicillin/streptomycin (GIBCO), and were maintained at 37 °C in a humid atmosphere with 5% CO_2_.

### 4.3. Western Blots

Protein extracts (50 µg total protein) were separated by sodium dodecyl sulfate polyacrylamide gel electrophoresis (SDS-PAGE) on 12% or 8% gels as indicated. Blots were blocked with 5% milk in 0.1% Tween-tris buffered saline (TBS) for anti-actin (1:5000), anti-CAV1 (1:3000), or 7% milk in 0.1% Tween-PBS for anti-AT2R (1:1000) or 5% serum in 0.1% Tween-PBS for anti-(pY14)CAV1 (1:3000) antibodies. Bound antibodies were detected with peroxidase-conjugated, anti-rabbit, or mouse IgG secondary antibodies. Protein levels were quantified by scanning densitometry, standardized to actin levels in the same samples (mean ± standard deviation, from three different experiments).

### 4.4. Transwell Migration Assays

Transwell assays were performed as described [16]. In this case, B16F10 or MDA-MB-231 cells were pretreated with 1 μM CGP42112 or vehicle (water) before seeding in transwell chambers.

### 4.5. Transendothelial Migration Assay (TEM)

Transendothelial migration assays were performed as described [17]. In this case, B16F10 or A375 cells were pretreated with 1 μM CGP42112 or vehicle (water) before addition to a monolayer of endothelial cells. Ten fields were quantified per condition.

### 4.6. Silencing of AT2R by shRNA

B16F10(cav-1) cells (500,000) were seeded in 60 mm plates and then transfected with a mixture of 5 pLKO.1 vectors encoding 5 different shRNAs against the AT2R mRNA (MISSION® shRNA Bacterial Glycerol Stock, SHCLNG-NM_007429; Sigma Aldrich, St. Louis, MO, USA ) or with the scrambled control sequence (Plasmid # 1864addGene). After a week of puromycin selection (5 μg/mL), clones of (B16F10(cav-1/sh-AT2R) and B16F10(cav-1/sh-scrmbl) were isolated, and the expression of AT2R was evaluated by western blotting.

### 4.7. AT2R Expression by Adenoviral Transduction

Adenoviruses encoding AT2R (Ad-NHA-AT2R) are bicistronic vectors that co-express both the angiotensin receptors and the green fluorescent protein (GFP) [24]. As a control, cells were transduced with the Ad ctrl adenovirus that drove the expression of the GFP protein under the control of the same promoter. MDA-MB-231 cells were transduced with adenovirus 48 h after plating with a multiplicity of infection (MOI) of 500,000 for AdAT2R and 25,000 for Ad GFP. 

### 4.8. Rab5-GTP and Rac1-GTP Pull-Down Assays

Rab5-GTP and Rac1-GTP pull-down assays were performed as previously described [15,16,17]. 

### 4.9. Metastasis Assays 

C57BL/6 mice were injected intravenously with either 200,000 B16F10(mock), B16F10 (cav-1), (B16F10 (cav-1/sh-AT2R), or (B16F10 (cav-1/sh-scrmbl) cells after CGP42112 or vehicle treatment. After 21 d, mice were sacrificed, and lungs were fixed in Fekete’s solution. Black tissue, corresponding to lung metastases, was separated from the rest of the lungs and weighed. Metastasis is expressed as black tissue mass/total lung mass (%) [22]. This study was performed according to the rules and standards established by the Bioethics Committee on Animal Research at the Facultad de Medicina, Universidad de Chile (CBA # 0897 FMUCH, date of approval: 23rd November 2016).

### 4.10. Measurement of PTP1B Activity 

The cells were washed in PBS supplemented with inhibitors of proteases and phosphatases (1 mM phenylmethylsulfonyl fluoride (PMSF), 1 mM Na3VO4, 10 mM NaF, 100 μg/mL benzamide, 10 μg/mL antipain, and 12.5 μg/mL leupeptin) on ice and subsequently lysed for 15 min in 20 mM Tris (pH 7.4), 150 mM NaCl, and 1% NP40 containing the inhibitor cocktail described above. The resulting cell homogenate was centrifuged at 16,000× *g* for 2 min, and the protein concentration of the supernatant was determined by the bicinchoninic acid (BCA) method. At least 2 g of proteins were immunoprecipitated with 2.5 μg of primary antibody (polyclonal anti-PTP1B) for 3 h at 4 °C on a rotating shaker. Then, 25 μL of a 1:1 mixture of protein A/G agarose was added and incubated for another 12 h at 4 °C. The suspension was then centrifuged at 1500× *g* for 5 min, and the pellet was resuspended in 50 μL acetate buffer at pH 5.6. Then, 5 μL of p-nitrophenyl phosphate was added as substrate, the mixture was incubated at 37 °C, and optical density at 415 nm was measured at different time points in a spectrophotometer.

### 4.11. Statistical Analysis 

Results were compared using one-way ANOVA with a Tukey’s post-test for multiple comparisons (graphic data) and unpaired *t* tests for pairwise comparisons (western blot data) using GraphPad Prism 5 software (San Diego, CA, USA). Values averaged from at least three independent experiments were compared. A *p* value < 0.05 was considered significant

## 5. Conclusions

Our findings identify a novel pathway by which AT2R reduces migration, invasion, and metastasis of melanoma and breast cancer cells. In doing so, novel therapeutic alternatives may become available to treat cancer metastasis.

## Figures and Tables

**Figure 1 cancers-11-01299-f001:**
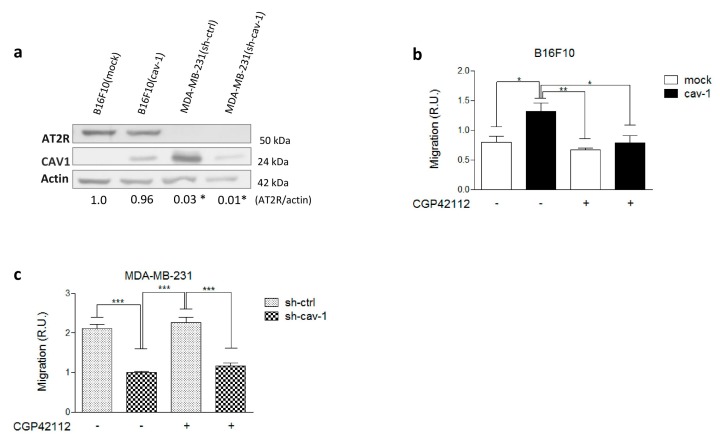
B16F10 cells express angiotensin II type-2 receptor (AT2R), and its activation decreases migration while AT2R is undetectable in MDA-MB-231. (**a**) Immunoblots of B16F10 melanoma and MDA-MB-231 cell lysates with antibodies against AT2R, caveolin-1 (CAV1), and β-actin. The numbers below individual lanes show the averaged AT2R/actin ratios normalized to B16F10(mock) (N = 3). B16F10(mock) (1.0 ± 0.04), B16F10(cav-1) (0.96 ± 0.22), MDA-MB-231(sh-ctrl) (0.03 ± 0.01), and MDA-MB-231(sh-cav-1) (0.01 ± 0.01). Migration assay with B16F10 (**b**) and MDA-MB-231 (**c**) cells were performed as described. Both cell lines were pretreated for 30 min with the AT2R agonist N-α−Nicotinoyl-Tyr-Lys-(N-α-Z-Arg)-His-Pro-Ile (CGP42112). The graphs show values normalized to the average of the control condition (N = 3) (mean ± standard error of mean). Results obtained were compared statistically as explained in Materials and Methods. Significant differences are indicated as *** *p* ≤ 0.001, ** *p* ≤ 0.01, and * *p* ≤ 0.05. (R.U., relative units).

**Figure 2 cancers-11-01299-f002:**
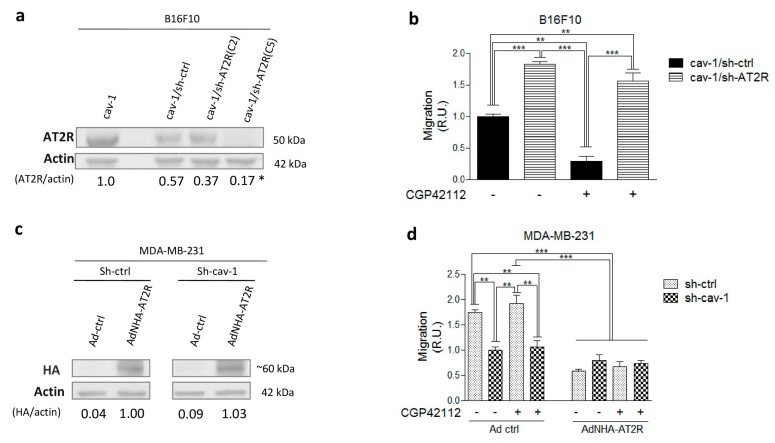
Depletion of angiotensin II type-2 receptor (AT2R) enhances B16F10(cav-1) cell migration, while AT2R overexpression abrogates MDA-MB-231 cell migration. (**a**) B16F10(cav-1) cells were transfected with a short hairpin AT2R (shAT2R) RNA-encoding vector to obtain B16F10(cav-1/sh-AT2R) cells. The clones with less AT2R expression (clon 2) and (clon 5) were compared with B16F10(cav-1/sh-ctrl) and B16F10(cav-1) by immunoblotting using antibodies against AT2R, caveolin-1 (CAV1), and β-actin. The numbers below individual lanes show the average values for AT2R/actin ratios normalized to B16F10(cav-1) (N = 3). B16F10(cav-1) (1.0 ± 0.16), B16F10(cav-1/sh-ctrl) (0.57 ± 0.08), B16F10(cav-1/sh-AT2R) clon 2 (C2) (0.37 ± 0.05), and B16F10(cav-1/sh-AT2R) clon 5 (C5) (0.17 ± 0.02). (**b**) Migration of B16F10(cav-1/sh -AT2R) clon 5 pretreated for 30 min with the AT2R agonist N-α-Nicotinoyl-Tyr-Lys-(N-α-Z-Arg)-His-Pro-Ile (CGP42112; 1 μM) was evaluated as described. (**c**) MDA-MB-231 cells transduced with AdNHA-AT2R or Ad-ctrl adenoviral were analyzed by immunoblotting with antibodies against the hemagglutinin (HA) tag and β-actin (N = 1). (**d**) Migration of transduced MDA-MB-231, pretreated for 30 min with CGP42112 (1 μM), was evaluated as described. The graphs show values normalized to the average of the control condition (N = 3) (mean ± standard error of mean). Results obtained were compared statistically as explained in Materials and Methods. Significant differences are indicated as *** *p* ≤ 0.001, ** *p* ≤ 0.01, and * *p* ≤ 0.05. (R.U., relative units).

**Figure 3 cancers-11-01299-f003:**
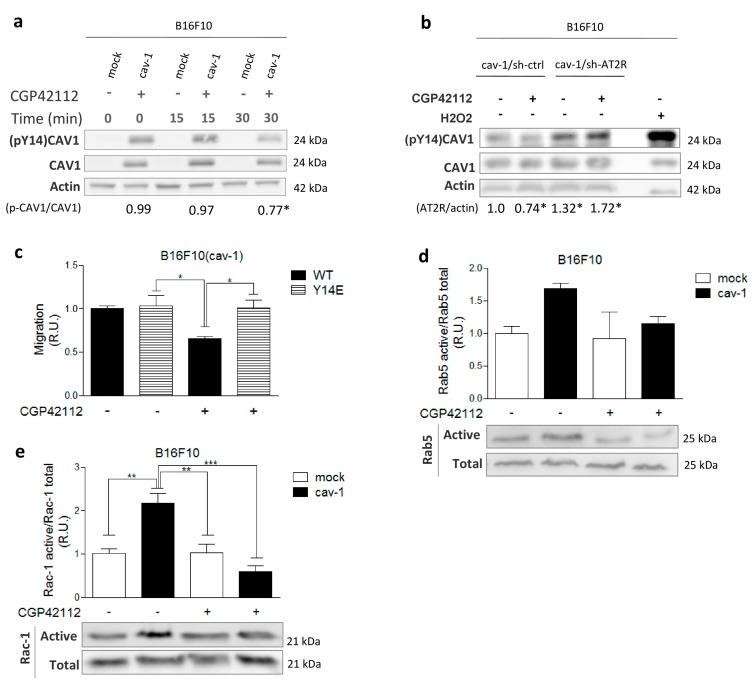
Angiotensin II type-2 receptor (AT2R) induces dephosphorylation of tyrosine-14 phosphorylated caveolin-1 (pY14(CAV1)) in B16F10(cav-1) cells and decreases Ras-related protein 5A (Rab5) and Ras-related C3 botulinum toxin substrate 1 (Rac1) activity. (**a**) Extracts from B16F10 cells pretreated for 0, 15, and 30 min with N-α−Nicotinoyl-Tyr-Lys-(N-α-Z-Arg)-His-Pro-Ile (CGP42112; 1 μM) were analyzed by western blotting using antibodies against pY14-CAV1, CAV1, and β-actin. The numbers below individual lanes indicate the average pY14-CAV1/CAV1 ratio normalized to the control condition (time 0) (N = 3). Time 0 (0.99 ± 0.09), 15 (0.97 ± 0.06), and 30 (0.77 ± 0.03) min. (**b**) B16F10(cav-1/Y14E) were pretreated for 30 min with CGP42112 (1 μM), and migration was evaluated as described. Values shown were normalized to those control B16F10(cav-1) cells without treatment (N = 3). (**c**) B16F10 cells were pretreated with CGP42112 (1 μM) for 30 min before evaluating small G-protein activity in pull-down assays (PD). Samples from PD and cell lysates were immunoblotted with antibodies against Rab5 (**d**), Rac-1 (**e**), and β-actin. Graphs show either the active Rab5/Total Rab5 (**d**) or active Rac-1/total Rac-1 (**e**) ratio normalized to control conditions (mean ± standard error of mean, N = 3). Results obtained were compared statistically as explained in Materials and Methods. Significant differences compared with the untreated condition (basal) of control cells are indicated. *** *p* ≤ 0.001, ** *p* ≤ 0.01, * *p* ≤ 0.05. (R.U., relative units).

**Figure 4 cancers-11-01299-f004:**
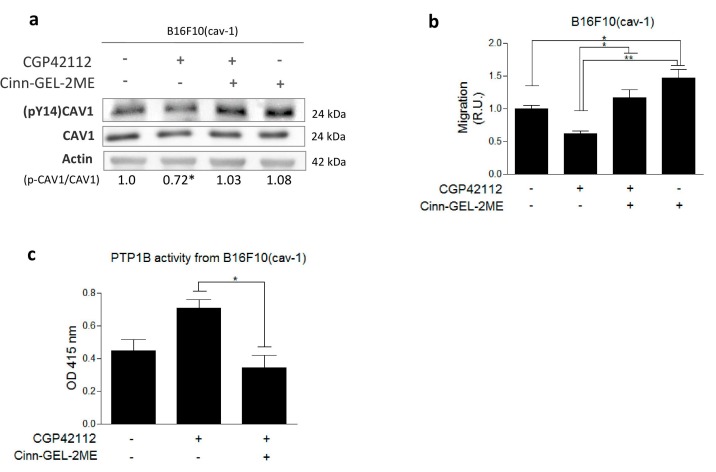
Angiotensin II type-2 receptor (AT2R) activation decreases non-receptor protein tyrosine phosphatase 1B (PTP1B) activity in B16F10(cav-1) cells. (**a**) Lysates of B16F10(cav-1) cells pretreated for 30 min with N-α−Nicotinoyl-Tyr-Lys-(N-α-Z-Arg)-His-Pro-Ile (CGP42112; 1 μM) and 12 h with the PTP1B inhibitor Cinn-GEL-2ME (5 μM) were analyzed by western blotting using antibodies specific for tyrosine-14 phosphorylated caveolin-1 (pY14-CAV1), CAV1, and β-actin. The numbers below individual lanes indicate the average pY14-CAV1/CAV1 ratios normalized to the control condition (without treatment) (N = 3). Vehicle (1.00 ± 0.05), CGP42112 (1 μM) (0.72 ± 0.07), CGP42112 + Cin-GEL-2ME (1.03 ± 0.16), and Cin-GEL-2ME (1.07 ± 0.20). (**b**) Migration of B16F10(cav-1) cells pretreated for 30 min with CGP42112 (1 μM) and/or Cinn-GEL-2ME (5 μM) for 12 h was evaluated as described. The graph shows values normalized to the average of the control condition (N = 3) (mean ± standard error of mean). (**c**) B16F10 cells were pre-incubated for 30 min with CGP42112 (1 μM) or Cinn-GEL-2ME (5 μM), and PTP1B activity was assayed as described. Changes in absorbance at 415 nm (OD 415 nm) indicative of phosphatase activity are shown. The graph shows values normalized to the average of the control condition (N = 3) (mean ± standard error of mean). Results obtained were compared statistically as explained in Materials and Methods. Significant differences are indicated as ** *p* ≤ 0.01, * *p* ≤ 0.05. (R.U., relative units)

**Figure 5 cancers-11-01299-f005:**
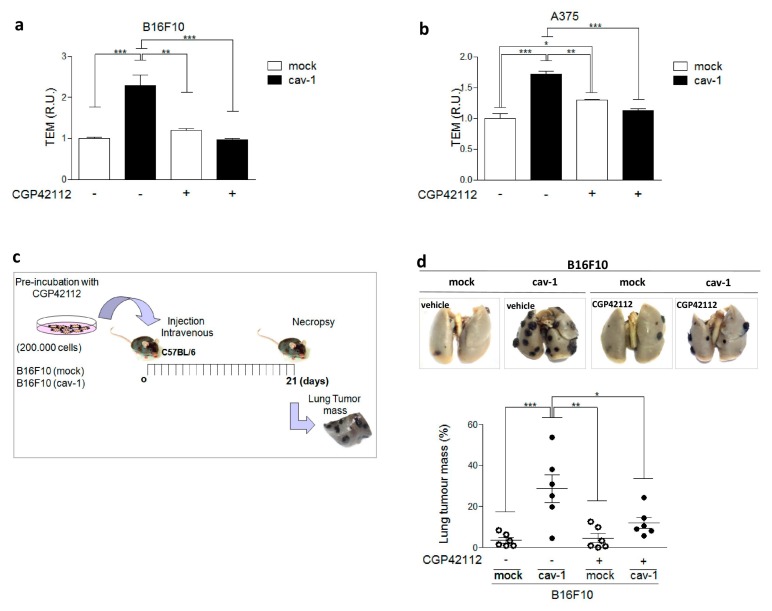
Angiotensin II type 2 receptor (AT2R) activation decreases transendothelial migration and metastasis of melanoma cells. (**a**) and (**b**) Transendothelial Migration (TEM) following pretreatment for 30 min with N-α−Nicotinoyl-Tyr-Lys-(N-α-Z-Arg)-His-Pro-Ile (CGP42112; 1 μM) or in controls was performed using B16F10 or A375 cells, respectively, expressing or not caveolin-1 (CAV1). Values obtained were normalized to the average values obtained for controls without treatment (N = 3) (mean ± standard error of mean). (**c**) Scheme illustrating the sequence of events in metastasis assays. For metastasis assays, B16F10(cav-1) or B16F10(mock) cells were pretreated with CGP42112 (1 μM) for 30 min. Metastasis in C57BL/6 mice was evaluated as described. (**d**) Representative images of lungs showing metastasis 21 d after inoculating the mice with the cells are shown in the upper panel. The graph shows a quantitative analysis of lung metastasis obtained by expressing lung tumor mass as a percentage of total lung mass. Data represent the (mean ± standard deviation) of three independent experiments with a total of 8 mice per condition. Results obtained were compared statistically as explained in Materials and Methods. Significant differences are indicated as *** *p* ≤ 0.001, ** *p* ≤ 0.01, * *p* ≤ 0.05. (R.U., relative units)

**Figure 6 cancers-11-01299-f006:**
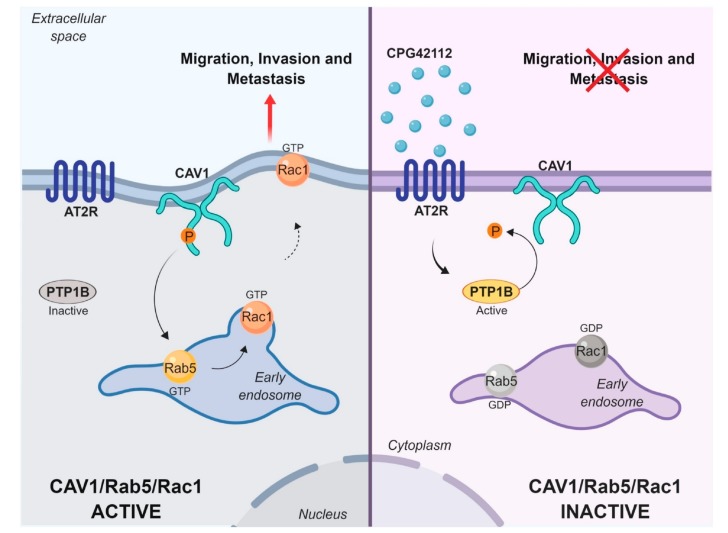
Model showing mechanism by which angiotensin II type 2 receptor (AT2R) inhibits the activity of the caveolin-1 (CAV1)/ Ras-related protein 5A (Rab5)/Ras-related C3 botulinum toxin substrate 1 (Rac1) pathway. CAV1 phosphorylated on tyrosine-14 (Y14) promotes Rab5 and then Rac-1 activation to increase migration, invasion, and metastasis. Non-receptor protein tyrosine phosphatase 1B (PTP1B) is a phosphatase that is part of a multiprotein complex with CAV1 and dephosphorylates CAV1 at Y14. Activation of AT2R with the agonist CGP42112 stimulates PTP1B activity and dephosphorylates CAV1, which in turn shuts down the CAV1/Rab5/Rac-1 signaling pathway, thereby effectively inhibiting CAV1-enhanced migration, invasion, and metastasis.

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
