# Peer review of "AT2 Receptor Mediated Activation of the Tyrosine Phosphatase PTP1B Blocks Caveolin-1 Enhanced Migration, Invasion and Metastasis of Cancer Cells"

_cancers, 2019, doi:10.3390/cancers11091299_

Round 1

Reviewer 1 Report

The manuscript by Martinez et al shows that activation of AT2R decreases CAV1-enhanced melanoma and breast cancer cell migration. Using a synthetic agonist, the authors further go on to show that AT2R inhibits cell migration by activating the tyrosine phosphatase PTP1B, dephosphorylating CAV1 and inhibiting the CAV1/Rab5/Rac-1 signaling axis. The animal study nicely validated cell culture experiments and overall, the study was done well. However, there are some major concerns and minor issues that will need to be addressed before the manuscript can be considered for publication.

Major concerns:

1.     While the authors did a nice job of using two different cell lines where levels of caveolin-1 were altered, the study would have benefited significantly by using an additional agonist instead of using a synthetic compound CGP42112 as a sole agonist for AT2R.

2.     The authors used shRNA against AT2R and adenovirus to overexpress AT2R in Fig 2 but only cell migration was assessed. The authors should complement findings regarding cav-1 phosphorylation, and Rab5/Rac-1 and PTP1B activities (Figs 3 &4) by altering AT2R expression.

3.     Similar to the first point, only one PTP1B inhibitor, Cinn-GEL2-ME, was used to assess the role of PTP1B. The authors should consider depleting PTP1B to complement observations from Fig 4.

4.     Also, while regulation of cav-1 phosphorylation was examined by studying the role of PTP1B, the role of kinases should also be examined. Non-receptor tyrosine kinases, like Src, Fyn, Yes and c-Abl that are known to phosphorylate cav-1 as mentioned in the manuscript should be assessed in four cell lines tested in the presence or absence of an AT2R agonist.

5.     For the observation of AT2R overexpression decreasing migration in the absence of CGP42112 (Fig 2d), the authors proposed the possibility of AT2R homo-dimerization. To study this effect in more detail, it would be helpful to determine phosphorylated cav-1 (Y14) levels in AT2R-overexpressing MDA-MB-231 cells in the presence and absence of CGP42112. This would also partly address my second concern. Also, for Fig 2c, the authors should show HA and actin bands from sh-ctrl and sh-cav-1 samples without cropping them separately so that readers can compare overexpressed AT2R levels in those two cell lines.

6.     In the introduction, the authors state that AT2R is overexpressed in cancer. Yet, studies in colorectal, breast and prostate cancers as well as Fig 5 show that AT2R suppresses metastasis. It would be helpful if the authors can explain this conundrum in the discussion section.

Minor issues:

1.     For Figs 1a, 3c, and 3d, levels of actin are not equal.

2.     For Fig 3b, the authors should consider relabeling the graph “B16F10(cav-1)” to be consistent with Fig 4b. The figure legend of Fig 3b can then be “WT” and “Y14E”.

3.     For Fig 3b, expression levels of cav-1 vs cav-1/Y14E should also be shown.

4.     For migration assays, analyses of statistical significance should be made between CGP44112 treated vs untreated cells. If they were done but no statistical significance was observed, the authors should state that in the figure legend.

5.     Related to the point #4, in line 139, the authors mention that “AT2R overexpression only decreased migration of cells that expressed endogenously CAV1 (MDA-MB-231 sh-ctrl), but not of the CAV1 silenced cells (MDA-MB-231 sh-cav-1).” However, there’s no indication from the figure that statistical analyses were performed between control and AT2R overexpressing CAV1 silenced cells. Therefore, p values should be shown for the cav1 silenced cells in the figure.

6.     For Fig 4c, to make a claim that phosphatase activity “was higher in CAV1 IPs from B16F10(cav-1) cells stimulated with CGP42112 and that this increase was blocked if cells were treated with the inhibitor Cinn-GEL2-ME,” the authors need to have a condition where cells were co-treated with both CGP42112 and Cinn-GEL2-ME.

7.     The authors should be consistent in presenting data for migration assays. For ex., Figs 1 and 3c & d are arranged based on the CGP4112 treatment whereas Figs 2 and 3b are grouped based on cell types.

Reviewer 2 Report

The authors demonstrate convincingly that AT2R plays a surprising role in regulating cancer metastasis. They do a thorough job fleshing out the mechanism of a likely signaling pathway leading to these effects. I only have minor recommendations. 

Figure 4c:

Based on the description in the text (line 202), “this increase was blocked…”, I expected that an experimental condition including treatment with bothCGP42112 and Cinn-GEL-2ME was performed. If it was not, then it should be done to correspond to the “rescue” experiments performed in Figures 4a and 4b. Or if the graph was simply mislabeled, then it should be corrected. 

Figure 4d:

The order of the images and bars should be changed to be consistent with Fig. 1b, 3a, 5a and 5b. This would make interpretation easier. 

Lines 304-305: Is there something in the medium that could be stimulating AT2R in the absence of the synthetic agonist?

Methods 4.11:

Since most data include more than two comparisons, ANOVA would be a more appropriate statistical analysis. A post-test can be used to make comparisons between individual conditions. GraphPad Prism 5 should be able to do this. 

Additional notes:

Line 61: “y” should be “and”

Lines 81-82: introduce the cell lines here as being either melanoma or breast cancer

Line 160: “y” should be “and”

Figure 5a and 5b: Units (R.U. or U.R. are not consistent and not defined in paper)

Lines 338 and 343: The standard English abbreviation for fetal bovine serum is FBS.

Lines 323 and 348: An anti-AT1R antibody is mentioned, but I don’t believe it was used in this study. 

Round 2

Reviewer 1 Report

The authors addressed most of the concerns adequately and in the process, the revised manuscript has been improved. The manuscript is now recommended for publication.